# Influence of Ripening Stage and Cultivar on Physicochemical Properties and Antioxidant Compositions of Aronia Grown in South Korea

**DOI:** 10.3390/foods8120598

**Published:** 2019-11-20

**Authors:** Haejo Yang, Young-Jun Kim, Youngjae Shin

**Affiliations:** 1Department of Environmental Horticulture, Dankook University, Cheonan, Chungnam 31116, Korea; gowh1231@naver.com; 2Department of Food Science and Technology, Seoul National University of Science and Technology, Seoul 01811, Korea; 3Department of Food Engineering, Dankook University, Cheonan, Chungnam 31116, Korea

**Keywords:** aronia grown in Korea, antioxidant compounds, catechol, chlorogenic acid, anthocyanin, antioxidant activity

## Abstract

The present study investigated the fruits of aronia (*Aronia melanocarpa*) across different stages of maturity and analyzed their physicochemical properties, antioxidant compositions, and activities. The selected aronia cultivars (‘Viking’, ‘McKenzie’, and ‘Kingstar K1’ were categorized based on maturity into the immature stage (red tip), intermediary stage (red), and mature stage (dark purple). The key sugar components of aronia fruits were fructose, glucose, and sorbitol, while the main organic acid was found to be malic acid. The antioxidant content and activity of all three aronia cultivars showed significantly higher values for the red tip stage than the red or dark purple stages. However, the total anthocyanin content of aronia was the highest at the dark purple stage in three cultivars. The main polyphenols in aronia fruits were found to be catechol and chlorogenic acid, with a decreasing tendency as maturation progressed. As a result, the red tip stage of aronia fruits contains comparatively more abundant flavonoids, phenolic compounds and polyphenols than the dark purple stage, with higher antioxidant activity.

## 1. Introduction

In recent years, berries have received considerable attention due to their high content of flavonoids, anthocyanins, and polyphenols. Further, berries are known to have a very beneficial effect on human health [1]. Anthocyanin is a naturally occurring secondary metabolite in plants, and it is known to be effective in preventing obesity, diabetes, and cardiovascular disease and in improving cognitive function [2]. These effects are related to antioxidant activity, which has the same mechanism as that corresponding to free radical scavenging activity, inhibition of peroxidase, and vasodilation. The major substances showing such biological activity are phenolic compounds, including anthocyanins, flavonoids, tannin, organic acids and vitamins [3].

Aronia (genus *Aronia*) is a deciduous shrub and grows in the wild in the northern United States. It is classified into *Aronia melanocarpa* (Michx.) Ell., which is known as black chokeberry, *Aronia arbutifolia* (L.) Pers., which is known as red chokeberry, and *Aronia prunifolia*, which is a purple chokeberry bred by cross-fertilization of *A. melanocarpa* and *A. arbutifolia* [4]. *A. melanocarpa* has mainly been used as a raw material for natural pigments in the food and pharmaceutical industries. It is the most widely grown cultivar since juice, jam and wine production became available [5]. The most important commercial *A. melanocarpa* cultivars in Europe and the United States are ‘Aron’, ‘Nero’, ‘Viking’, ‘Hugin’ and ‘Rubina’. The main polyphenol components of ‘Viking’ fruit are proanthocyanidin and anthocyanin; therefore, the ‘Viking’ fruit has been widely used as a natural raw material for food, pharmaceuticals, and cosmetics.

It has also been known to have significant antioxidant, anti-inflammatory and anti-cancer effects, in addition to being effective in blood glucose control [6]. Therefore, in this study, the typical domestic *A. melanocarpa* cultivars, ‘Viking’, ‘McKenzie’ and ‘Kingstar K1’ were selected and they were harvested at the red tip, red, and dark purple stages of fruit maturity. The objective of this study is to determine changes in the physicochemical properties, antioxidant profiles and activities of aronia and to investigate the effects of cultivar and maturity stage at harvest because none of the studies has considered these two factors simultaneously.

## 2. Materials and Methods 

### 2.1. Plant Materials

The aronia samples were *A. melanocarpa* ‘Viking’, ‘McKenzie’, and ‘Kingstar K1’ grown in a farm in Cheongju, Chungcheongbuk-do, South Korea. The samples were collected from 30 bushes on each cultivar, depending on the maturity stage. There was a total of 27 samples since there were 3 cultivars × 3 maturities × 3 replicates. We collected 3 kg for each sample. The changes in physicochemical characteristics, antioxidant compound composition and antioxidant activity of the samples were analyzed according to maturity. Fruit maturity was classified into three stages according to the degree of fruit coloring: red tip, red, and dark purple. The fruit at these three stages was harvested in mid-June of 2017 (red tip), mid-July of 2017 (red), and in mid-August of 2017 (dark purple), respectively. The climatic condition data of mean temperature (°C) and total precipitation (mm) were collected from the Korea Meteorological Administration. The average temperature (°C) and total precipitation (mm) of each harvest month in 2017 were 23.4 °C and 17.5 mm, 27.1 °C and 789.1 mm, and 26.1 °C and 225.2 mm, respectively.

The harvested aronia fruits were immediately transferred to the laboratory, and the damaged fruits were removed. Then, the fruits were rapidly frozen with liquid nitrogen and then extracted for analysis of antioxidant compounds and antioxidant activities. The physicochemical qualities of the fruits, such as moisture content, color, firmness, pH and titratable acidity were analyzed on the day of the harvest. Photographs were also taken under the same conditions using a digital camera (EOS 550D, Canon, Tokyo, Japan) for visual comparison for color analysis.

### 2.2. Chemical Reagents 

Ethanol, Folin–Ciocalteu’s phenol reagent, 2,2-diphenyl-1-picrylhydrazyl (DPPH), 2,2-azino-bis (3-ethylbenzothiazoline-6-sulfonic acid) (ABTS), aluminum chloride hexahydrate, sodium nitrite, (+)-catechin, gallic acid monohydrate, sodium acetate anhydrous, potassium chloride, hydrochloric acid, sodium carbonate anhydrous, oxlalic acid, tartaric acid, malic acid, lactate, acetic acid, citric acid, succinic acid, fumaric acid, fructose, glucose, sucrose, maltose, lactose, sorbitol, gallic acid, protocatechuic acid, catechol, catechin, chlorogenic acid, epigallocatechin gallate, caffeic acid, epicatechin, syringic acid, 4-methycatechol, epicatechin gallate, p-coumaric acid, ferulic acid, and rutin were purchased from Sigma (St. Louis, MO, USA).

### 2.3. Extraction for Measurement of Antioxidant Compounds and Activities

To prepare the fruit extract, approximately 40 g of frozen aronia fruit was ground to a fine powder under liquid nitrogen by cold mortar and pestle and 30 g of the resultant powder was added to 300 mL of 80% ethanol, and then this mixture was homogenized in a commercial blender (HR-2171, Phillips, Seoul, Korea) three times for 3 min each time. The homogenate was centrifuged at 5230× *g* for 20 min using a centrifuge (Mega 21R, Hanil, Incheon, Korea). The supernatant was filtered using a Whatman #2 filter paper and concentrated using a rotary evaporator (N-1000, Eyela, Tokyo, Japan). The samples were then brought to 30 mL with deionized water, expressed as mg of fruit on a fresh weight basis, divided into several aliquots, and kept frozen at −20 °C until analysis.

### 2.4. Color Analysis

The color change of the fruits was represented as L* value (lightness ranging from 0 = black to 100 = white), a* value (redness), and b* value (yellowness) using a colorimeter (Chroma meter CR-400, Minolta, Tokyo, Japan). The Chroma Meter was calibrated regularly using the white calibration plate (Y = 87.8, x = 0.3156, y = 0.3229). For each measurement, 30 fruits were measured three times, and the mean value was calculated. The readings were taken around the equatorial region of each fruit.

### 2.5. Soluble Solid Content, Firmness, Titratable Acidity, pH and Moisture Content

The firmness of the fruit was measured using a fruit hardness tester (FHM-1, Dementra Co., Ltd., Tokyo, Japan). The intact skin part of the fruit was penetrated using a 12Φ × 10 mm cone probe, and the value was expressed in newtons (N). The soluble solid content, titratable acidity, and pH were measured using a mixture of 10 g of fruit and 100 mL of deionized water that was homogenized with an Oster blender (Oster, Milwaukee, WI, USA). Soluble solid content was measured using a digital refractometer (PAL-1, Atago Co, Ltd., Tokyo, Japan). Titratable acidity was determined by mixing 2 g of the mixture and 200 mL of distilled water and performing a neutralization titration with 0.1 N NaOH and 3–4 drops of 1% phenolphthalein solution. The pH was measured using a pH meter (Starter300, Ohaus Co, Ltd., Parsippany, NJ, USA). To measure the water content of the fruit, 20 g of fruit was weighed and stored in an incubator (IB-05G, Jeio Tech Co., Ltd., Seoul, Korea) set to 60 °C until there was no change in weight. Then, fruits were weighed again, and weight loss percentage was obtained.

### 2.6. Organic Acid Compositions

Organic acid composition profiling was performed as described by Kim and Shin [7], with some modifications. An Agilent 1100 series (Agilent Technologies, Wilmington, DE, USA) with a diode array detector (DAD) was used for the individual organic acid analysis. The extract was diluted in ten-fold distilled water and the samples were filtered through a 0.45 μm syringe. A prevail organic acid column (250 × 4.6 mm i.d., 5 μm, Hichrom Ltd., Reading, UK) was used at 25 °C, and a DAD was positioned at 210 nm. The HPLC (High-Performance Liquid Chromatography) mobile phase was 25 mM KH_2_PO_4_ adjusted to pH 2.1 using H_3_PO_4_ with a 1.0 mL/min flow rate, and the injection volume was 3 μL. For the calibration curves, the three different points (4, 20, 100 mg/100 g for oxlate, tartarate, malate, lactate, acetate, citrate, succinate, and fumarate) were prepared with standard solutions, and the unit for the results was expressed in mg/100 g of fresh weight (FW).

### 2.7. Sugar Composition

The sugar composition profiling was performed as described by Kim and Shin [7]. Aronia sample extracts were diluted ten-fold in distilled water and then the samples were filtered through a 0.45 μm syringe filter. HPLC analyses were carried out using an UltiMate 3000 HPLC system (Thermo Fisher Scientific, Waltham, MA, USA) with a Refractomax 520 refractive index (RI) detector (ERC Inc, Saitama, Japan). A carbohydrate high-performance column (250 × 4.6 mm i.d., 4 μm, Waters Corp., Milford, MA, USA) was utilized at 30 °C for individual sugar identification. The mobile phase was 79% acetonitrile in distilled water with a 1.0 mL/min flow rate, and the injection volume was 10 μL. Standard calibration curves were prepared with 200, 500 and 1000 mg/100 g of fructose, glucose, sucrose, maltose, lactose and sorbitol as reference materials. The determination was performed three times for each sample, and the results were shown as mg/100 g of FW.

### 2.8. Total Anthocyanin Analysis

The total anthocyanin content of fruit extracts was measured by the pH differential method [8,9]. The extracts were mixed with 0.025 M potassium chloride buffer (pH 1.0) and 0.4 M sodium acetate buffer (pH 4.5), and absorbance was measured at 510 and 700 nm using a spectrophotometer (Optizen POP, Mecasys, Daejeon, Korea). The total anthocyanin content was calculated using the following equation and expressed as mg cyanidin 3-glucoside equivalents (CGE)/100 g FW.
(1)$$\mathrm{Total}\;\mathrm{anthocyanin}\;\mathrm{content}\;(\mathrm{mg}\;\mathrm{CGE}/100\;g\;\mathrm{FW})\;=\;\frac{\left(A\;\times \;\mathrm{MW}\;\times \;D\;\times \;1000\right)}{\varepsilon }$$
where MW (molecular weight of cyanidin 3-glucoside) = 449.2; D = dilution factor; ε (molar extinction coefficient of cyanidin 3-glucoside) = 26,900.
A (absorbance value) = ((A510 nm − A700 nm)_pH 1.0_ − (A510 nm − A700 nm)_pH 4.5_)(2)

### 2.9. Total Flavonoid Analysis

The total flavonoid content of fruit extracts was measured by the colorimetric assay method [8,9]. In total, 0.3 mL of 5% NaNO_2_ was added to a 1 mL aliquot of the diluted sample mixed with deionized water, thoroughly mixed, and allowed to react for 5 min. Then, 0. 3 mL of 10% AlCl_3_ was added, thoroughly mixed, and allowed to react for 6 min. Lastly, 2 mL of 1 N NaOH and 2.4 mL of distilled water were added to adjust the total volume to 10 mL. Then, the absorbance was measured at a wavelength of 510 nm using a spectrophotometer. The standard curve was measured using catechin, and the total flavonoid content was expressed as mg catechin equivalents (CE)/100 g FW.

### 2.10. Total Phenolic Analysis

Total phenolic content of fruit extracts was measured by the Folin–Ciocalteu colorimetric method [8,9]. 0.2 mL of the Folin–Ciocalteu reagent was added to a 0.2 mL aliquot of the diluted sample mixed with deionized water, thoroughly mixed, and allowed to stand at room temperature for 6 min. Then, 2 mL of 7% NaCO_3_ was added and allowed to stand for 90 min at room temperature in a dark place, and absorbance was measured at 750 nm wavelength using a spectrophotometer. The standard curve was measured using gallic acid, and the total phenolic compound content was expressed as mg gallic acid equivalents (GAE)/100 FW.

### 2.11. Polyphenol Analysis

The polyphenol content of fruit extracts was analyzed using a modified method by Chen [10]. The extract was diluted twenty times with diluted solution (KH_2_PO_4_:MeOH:D.W = 2:3:15), and the diluted solution was filtered with a 0.45 µm syringe filter. The filtered samples were analyzed using HPLC (Thermo Fisher UltiMate 3000, Thermo Scientific, Bremen, Germany), and Eclipse XDBC-18 column (150 × 4.6 mm, 5 μm, Agilent Technologies, Wilmington, DE, USA) at 40 °C. Acetic acid (3%) was used as the HPLC mobile phase, and the detector was set to a wavelength of 280 nm and a flow rate of 1.0 mL/min, where 10 μL of the samples were injected and analyzed. The standard calibration curve was measured using the following as standards: gallic acid, protocatechuic acid, catechol, catechin, chlorogenic acid, epigallocatechin gallate, caffeic acid, epicatechin, syringic acid, 4-methycatechol, epicatechin gallate, p-coumaric acid, ferulic acid, and rutin. The content was expressed as mg/100 g FW.

### 2.12. DPPH Radical Scavenging Activity Analysis

The DPPH radical scavenging activity was measured using a modification of the method described by Brand-Williams [11]. The DPPH solution was prepared by dissolving the 0.00789 g of powdered DPPH into 200 mL of methanol. The test solution containing 50 μL of the diluted solution mixed with deionized water and 2950 μL of 0.2 mM DPPH solution was allowed to react for 30 min, and the absorbance was measured at a wavelength of 517 nm using a spectrophotometer. The antioxidant activity of the extract was expressed as mg vitamin C equivalents (VCE)/100 g FW.

### 2.13. ABTS Radical Scavenging Activity Analysis

The ABTS radical scavenging activity of the extracted sample was measured using ABTS radicals [12]. The test solution containing 20 μL of the diluted solution mixed with deionized water and 980 μL of the ABTS reaction solution was reacted at 37 °C for 10 min, and the absorbance was measured at a wavelength of 734 nm using a spectrophotometer. The antioxidant activity of the extract was expressed as mg vitamin C equivalents (VCE)/100 g FW.

### 2.14. Statistical Analysis

For statistical analysis, one-way analysis of variance (ANOVA) was performed using SPSS 20 program (SPSS Inc., Chicago, IL, USA), and Duncan’s multiple range test was used for the significance of each average value (*p* < 0.05). The correlation between the mean values of each parameter was expressed using Pearson’s correlation coefficient. The data were expressed as the mean ± standard deviation from triplicate determination.

## 3. Results and Discussion

### 3.1. Color

The color and appearance at harvest day depending on the maturity of aronia are shown in Figure 1, and Hunter L, a, and b color of *A. melanocarpa* fruits at different maturity stages are shown in Table 1. In all three cultivars, lightness (L value) decreased as the fruit color darkened during the ripening process from red tip stage to dark purple stage. Redness (a value) increased gradually from red tip stage to red stage in all three cultivars but decreased at dark purple stage when the fruit color darkened. Yellowness (b value) showed a tendency to decrease from a positive value to a negative value in all three cultivars as maturation progressed.

### 3.2. Soluble Solid Content, Firmness, Titratable Acidity, pH, and Moisture Content

The soluble solid content, firmness, titratable acidity, pH, and sugar/acid ratio of aronia fruit are shown in Table 2. The soluble solid content significantly decreased from the red tip stage to the red stage and slightly increased from the red stage to the dark purple stage without any significant difference. Kaack and Kuhn [13] also reported that soluble solid content increased during the ripening process of black chokeberry. Firmness decreased with maturation in all three cultivars, and water content increased as maturation progressed. Taheri [14] demonstrated that differences in water content of fruits generally did not affect the polyphenol content and water content increased with maturation stage. The titratable acidity was 1.26%–1.30%, the differences between cultivars and between maturity stages were negligible, and the highest pH was shown at the dark purple stage. Many studies on the quality analysis of aronia fruit reveal that the soluble solid content of aronia fruit is in the range of 12% to 20%, and the titratable acidity is 0.7%–1.4% [15]. The results of this study showed that the sugar-acid ratios of the aronia fruit at the red and dark purple stages were not significantly different in three cultivars. In general, during the ripening process of fruits, the soluble solid content and sugar content are increased while titratable acidity and organic acid content are decreased [16]. However, physicochemical properties of the fruit are affected by various factors such as climatic conditions, soil composition, and maturity during harvest, and these factors are known to have a significant influence on fruit quality [17].

### 3.3. Quantification of Individual Organic Acids

Organic acids, sugars and volatile compounds, are known to contribute greatly to the taste and flavor of fruits [18]. Table 2 shows the results of measuring organic acid content of aronia fruit depending on maturity. The malic acid content of three cultivars increased significantly as the fruit became more mature. In comparison, between cultivars at each maturity stage, ‘Viking’ and ‘McKenzie’ showed similar malic acid content. However, the malic acid content of ‘Kingstar K1’ at the red and dark purple stage was significantly lower than that of other cultivars. Mikulic-Petkovsek et al. [19] reported that the major organic acids of berries were citric acid and malic acid. In this study, malic acid was found to be main organic acid, and other organic acids (oxlalic acid, tartaric acid, lactate, acetic acid, citric acid, succinic acid, fumaric acid) were not detected (below the limit of quantification (LOQ)). Therefore, malic acid was dominant at the dark purple stage where the fruit was completely ripe, which is consistent with the report that malic acid was the main organic acid of black chokeberry [15].

### 3.4. Quantification of Individual Sugars

Table 3 shows the sugar content of aronia fruit according to maturity. The contents of fructose, glucose, and sorbitol were significantly increased as ripening progressed. It was found that the sorbitol content was higher than fructose and glucose in each cultivar at all stages. Mikulic-Petkovsek et al. [19] showed that the sugar contents of aronia were glucose 3.54 g/100 g, fructose 2.82 g/100 g, sucrose 0.412 g/100 g, and sorbitol 4.62 g/100 g, but the bitter taste of ripened chokeberry was stronger than sweet taste because of its high content of polyphenol. Sorbitol was found to be the major sugar component of aronia fruit.

### 3.5. Total Anthocyanin Content

The total anthocyanin content of aronia fruit according to cultivar and maturity was the highest at the dark purple stage in three cultivars—224.62 mg/100 g FW in ‘Viking’; 226.33 mg/100 g FW in ‘McKenzie’; and 185.84 mg/100 g FW in ‘Kingstar K1’ (*p* < 0.05)—and was significantly higher than that at the red tip and red stages. The content was significantly higher in ‘Viking’ and ‘McKenzie’ than in ‘Kingstar K1’ (Figure 2A). In this study, the anthocyanin content of aronia fruit tended to increase with maturation. Siriwoharn [20] also reported a similar result that the total anthocyanin of blackberry ‘Marion’ increased from 74 mg/100 g FW in the unripe fruit to 317 mg/100 g FW in the ripe fruit, as maturation progressed and the total anthocyanin of ‘Evergreen’ also increased from 69.9 mg/100 g FW to 164 mg/100 g FW. According to Wang and Lin [21] on analysis of anthocyanin content according to green, pink, and ripe stage of berries, blackberry showed 0.9, 9.1, and 152.8 mg/100 g, respectively; black raspberry showed 1.7, 22.8, and 197.2 mg/100 g, respectively; red raspberry showed 1.0, 7.2, and 68.0 mg/100 g, respectively; strawberry showed 0.3, 5.5, and 31.9 mg/100 g, respectively. Thus, the anthocyanin content increased with maturation progress in all berries. The anthocyanin content of berries was higher in the order of black chokeberry, black currant, blackberry, and raspberry [22].

### 3.6. Total Flavonoid Content

The total flavonoid content of aronia fruit was highest at the red tip stage in three cultivars: 3175.52 mg/100 g FW in ‘Viking’; 3544.7 mg/100 g FW in ‘McKenzie’; and 3212.50 mg/100 g FW in ‘Kingstar K1’. Thus, the flavonoid content at the red tip stage was significantly higher than that at the red stage and dark purple stage (*p* < 0.05) (Figure 2B). In particular, the total flavonoid content of ‘McKenzie’ at the red tip stage was significantly higher than that of ‘Viking’ and ‘Kingstar K1’ (*p* < 0.05). At the red stage, the total flavonoid content of ‘Kingstar K1’ was significantly higher than that of ‘Viking’ and ‘McKenzie’. However, at the dark purple stage, there was no significant difference between the three cultivars. Therefore, the flavonoid content of aronia fruit decreased significantly as the ripening process progressed. Wang’s [23] study on the change of flavonoid content in red raspberry according to maturity reported that the content of quercetin and kaempferol decreased significantly during ripening.

### 3.7. Total Phenolics Content

The total phenolic content of aronia fruit was highest at the red tip stage in three cultivars: 3955.28 mg/100 g FW in ‘Viking’; 4393.50 mg/100 g FW in ‘McKenzie’; and 4284.55 mg/100 g FW in ‘Kingstar K1’. Thus, the total phenolic content at the red tip stage was significantly higher than that at the red and dark purple stage (*p* < 0.05) (Figure 2C). In the comparison between cultivars according to maturity, the phenolic content was higher at the red tip stage in the order of ‘McKenzie’, ‘Kingstar K1’, and ‘Viking’ (*p* < 0.05). At the red stage, ‘Viking’ and ‘Kingstar K1’ showed significantly higher phenolic contents than ‘McKenzie’ (*p* < 0.05), but the phenolic content was significantly higher at the dark purple stage in the order of ‘McKenzie’, ‘Viking’, and ‘Kingstar K1’ (*p* < 0.05). These results suggest that the total phenolic content of aronia decreased during the ripening process, which was a similar pattern to that of the flavonoid content.

Previous studies have shown that the anthocyanin content of blackberry increased significantly during ripening, but ellagitannin, flavonol, and ellagic acid content decreased significantly during the ripening process [24]. Black chokeberry is known to be a rich source of phenolic compounds, and polyphenol content was reported to be significantly higher in chokeberry (690.2 mg/100 g) than in red raspberry (177.5 mg/100 g) and blackberry (289.3 mg/100 g) [22,25]. In addition, the higher content of phenol compounds in unripe fruits than that in ripe fruits results from a large amount of phenylpropanoid substances contained in unripe fruits.

### 3.8. Polyphenol Content

The polyphenol content of 80% ethanol extract of aronia fruit was measured. As a result, major polyphenols were found to be catechol and chlorogenic acid (Table 4), and other polyphenol compounds (gallic acid, protocatechuic acid, catechin, epigallocatechin gallate, caffeic acid, epicatechin, syringic acid, 4-methycatechol, epicatechin gallate, p-coumaric acid, ferulic acid, and rutin) were not detected (below the LOQ). The catechol content of each cultivar at the red tip, red, and dark purple stage was 256.11, 134.18, and 95.05 mg/100 g, respectively, in ‘Viking’; 399.01, 118.02, and 81.80 mg/100 g, respectively, in ‘McKenzie’; and 289.03, 118.33, and 70.69 mg/100 g, respectively, in ‘Kingstar KI’. Thus, the catechol content was the highest at the unripe stage and decreased significantly as maturation progressed. The chlorogenic acid content at the red tip, red, and dark purple stage was 120.82, 72.31, and 64.69 mg/100 g, respectively, in ‘Viking’; 176.67, 68.75, and 56.72 mg/100 g, respectively, in ‘McKenzie’; and 132.58, 72.87, and 49.76 mg/100 g, respectively, in ‘Kingstar KI’. Thus, the chlorogenic acid content showed a similar pattern to the catechol content. Sidor and Gramza-Michałowska [26] reported that Chokeberries mostly contain chlorogenic and neochlorogenic acids. Other phenolic acids are cryptochlorogenic acid, p-coumaric acid and its derivatives, caffeic acid and its derivatives, protocatechuic, vanillic, ferulic, salicylic, syringic, 4-hydroxybenzoic and ellagic acids. According to Taheri [14], approximately 78% of the hydroxycinnamic acid content of aronia was chlorogenic acid, and all aronia cultivars contained chlorogenic acid at approximately 3.11–13.5 mg/g DW. Määttä-Riihinen [27] reported that the hydroxycinnamic acid content of chokeberry was detected at 0.892 mg/g FW, which is similar to the result of this study. Wang and Lin [21] reported that phenolic content in berries was affected by maturity stage at harvest, genetic differences, environmental conditions at harvest, post-harvest storage conditions and processing. According to Zheng and Wang’s [28] study on oxygen radical absorbance capacity of phenols in blueberry, cranberry, chokeberry and lingonberry, berries contained different contents of polyphenol depending on cultivar and genotype, and these polyphenols have high antioxidant activity. They also reported that higher polyphenol content leads to a higher antioxidant activity, which varies depending on the structure and composition of berries. In this study, the chlorogenic acid content in the unripe aronia was abundant, and chlorogenic acid, ferulic acid, caffeic acid, and p-coumaric acid are known to have high antibacterial activity [29]. Therefore, the results of this study suggest that the unripe aronia fruit containing the highest chlorogenic acid is useful in various food manufacturing fields.

### 3.9. DPPH Radical Scavenging Activity

The DPPH radical scavenging activity of aronia fruits was highest at the red tip stage in all three cultivars: 4141.44 mg/100 g FW in ‘Viking’; 4565.28 mg/100 g FW in ‘McKenzie’; and 4313.23 mg/100 g FW in ‘Kingstar K1’. Thus, the DPPH radical scavenging activity was significantly higher at the red tip stage than at the red and dark purple stages (*p* < 0.05) (Figure 3A). In the comparison between cultivars according to maturity, the DPPH radical scavenging activity was significantly higher at the red tip stage in the order of ‘McKenzie’, ‘Kingstar K1’, and ‘Viking’ (*p* < 0.05), and it was significantly higher at the red stage in the order of ‘Kingstar K1’, ‘Viking’, and ‘McKenzie’ (*p* < 0.05). At the dark purple stage, the DPPH radical scavenging activity was significantly higher in ‘Viking’ than in ‘McKenzie’ and ‘Kingstar K1’. In this study, the changes in DPPH radical scavenging activity according to maturity showed a pattern similar to the changes in the total phenolic content and total flavonoid content. Benvenuti et al. [22] studied the DPPH radical scavenging activity of blackberry, blackcurrant, chokeberry, raspberry. They indicated a relatively high potential of chokeberries and selected blackcurrant varieties.

### 3.10. ABTS Radical Scavenging Activity Analysis

The ABTS radical scavenging activity of aronia fruit was highest at the red tip stage in all three cultivars: 5976.86 mg/100 g FW in ‘Viking’; 6107.57 mg/100 g FW in ‘McKenzie’; 6090.15 mg/100 g FW in ‘Kingstar K1’. Thus, the ABTS radical scavenging activity at the red tip stage was significantly higher than that at the red and dark purple stage (*p* < 0.05) (Figure 3B). At the red stage, the ABTS radical scavenging activity was significantly higher in ‘Viking’ and ‘Kingstar K1’ than in ‘McKenzie’, and at the dark purple stage, there was no significant difference among the three cultivars. According to Wang and Lin’s [21] study, blackberry, raspberry and strawberry, which were harvested at green stage, showed higher antioxidant activity than those at pink and ripe stage, similar to the results of this study. Tarko et al. [30] also reported that chokeberry fruit components were the most active scavengers of the ABTS radical cation among chokeberries, apples, plums, pears, bananas and melons.

### 3.11. Correlation between Physicochemical Quality, Antioxidant Substances and Antioxidant Activity in Aronia Fruit

The correlation between antioxidant compound and antioxidant activity in aronia is shown in Table 5. The correlation between anthocyanin content and antioxidant compounds or activities were as follows: total flavonoid (*R* = −0.753), total phenolics (*R* = −0.705), DPPH radical scavenging activity (*R* = −0.780), ABTS radical scavenging activity (*R* = −0.839), catechol (*R* = −0.674), and chlorogenic acid (*R* = −0.655). These results indicate that the anthocyanin content of aronia increased during the ripening process, while the antioxidant compound, antioxidant activity and polyphenol content decreased during the ripening process. The correlation between total flavonoid and total phenolics was *R* = 0.991 and the correlation between total flavonoid and antioxidant activities were also high: DPPH radical scavenging activity *R* = 0.997 and ABTS radical scavenging activity (*R* = 0.979). Total phenolics and antioxidant activities were highly correlated and the correlation between total phenolics and catechol was *R* = 0.953; and the correlation between total phenolics and chlorogenic acid was *R* = 0.940. Wang and Lin [21] also found a negative correlation between the anthocyanin content of strawberry and ORAC (Oxygen radical absorbance capacity) value at green stage. Castrejón et al. [31] reported a strong correlation between antioxidant activity and total phenolic content in all blueberry cultivars. In addition, the remarkably higher total phenolic compound content and antioxidant activity in unripe fruits than in ripe fruits were due to a high concentration of hydroxycinnamic acid before ripening. Therefore, total flavonoid, total phenolics, and chlorogenic acid except total anthocyanin are excellent antioxidant compounds.

## 4. Conclusions

In summary, the physicochemical properties, changes in antioxidant compound composition, and antioxidant activity of three *A. melanocarpa* cultivars cultivated in Korea were analyzed in this study. It was found that the anthocyanin content of three *A. melanocarpa* cultivar fruits increased with maturation, but the contents of catechol, chlorogenic acid, total flavonoid, and total phenolics decreased as maturation progressed. In particular, catechol and chlorogenic acid were found to be higher in unripe fruits than in ripe fruits. Catechol, chlorogenic acid and total phenolics and antioxidant activity showed a very strong correlation. The major sugar components in aronia were identified as fructose, glucose, and sorbitol, and their concentrations increased as maturation progressed. Ripe fruits, which have been used as main raw materials of processed aronia food, had a higher anthocyanin content than unripe fruits, but polyphenol, antioxidant compound and antioxidant activity were higher in unripe fruits. However, there were no significant differences in physicochemical qualities, antioxidant compositions and activities among cultivars. This suggests that red tip aronia, unripe fruit, could be an effective source of health-beneficial antioxidant compounds as functional food ingredients. Thus, aronia fruits are expected to be useful in various fields other than processed food, and they may positively affect the farm market, thereby contributing to increasing farm household income.

## Figures and Tables

**Figure 1 foods-08-00598-f001:**
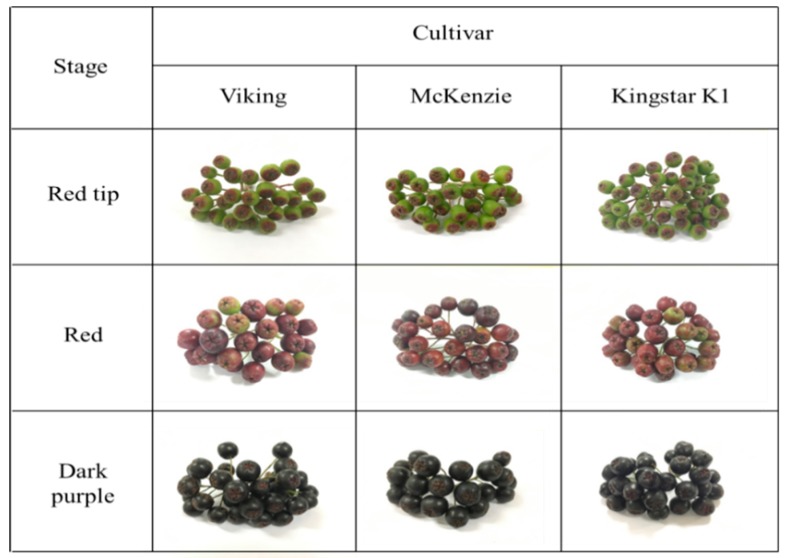
Color and appearance of *Aronia melanocarpa* fruits at different maturity stages.

**Figure 2 foods-08-00598-f002:**
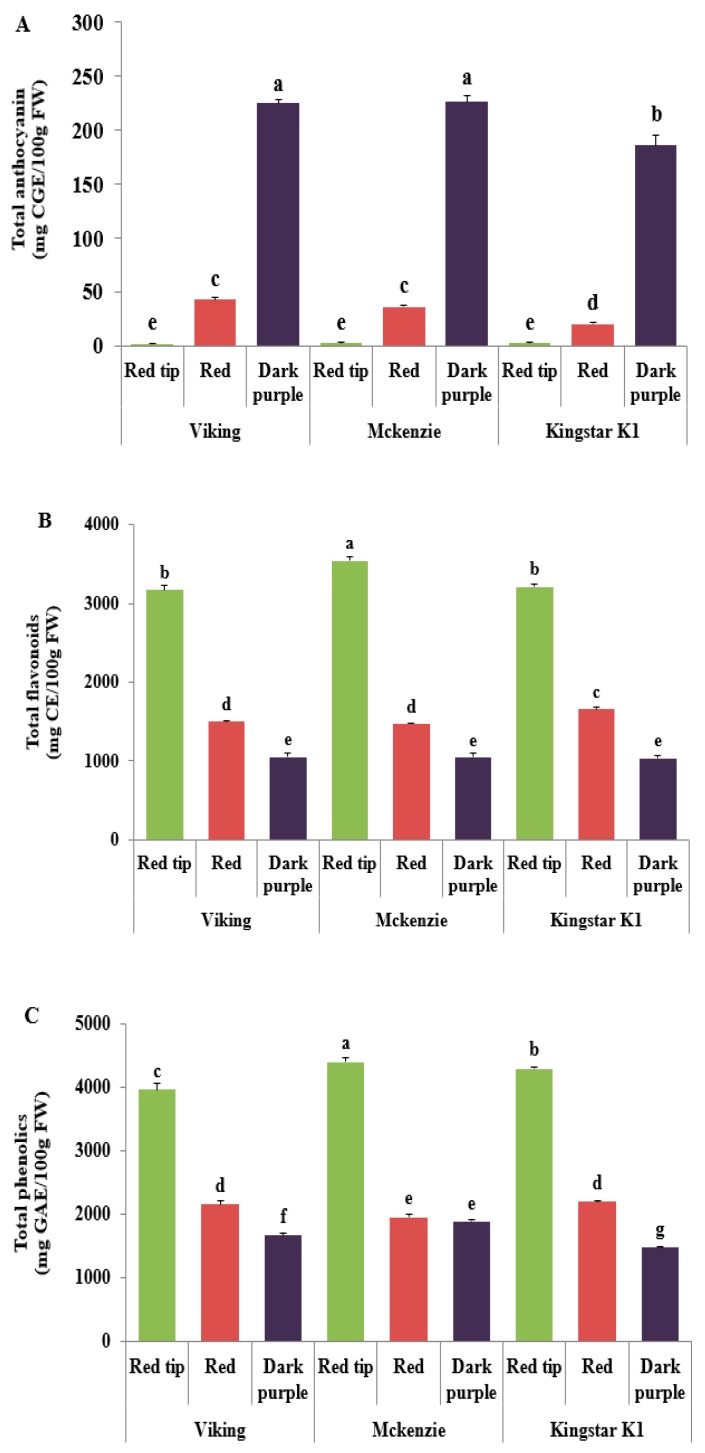
Total anthocyanin, total flavonoids and total phenolics contents of *Aronia melanocarpa* fruits at different maturity stages: (**A**) Total anthocyanin; (**B**) Total flavonoids; (**C**) Total phenolics. Vertical bars indicate standard deviation. Different letters are significant differences by Duncan’s multiple range test (*p* < 0.05).

**Figure 3 foods-08-00598-f003:**
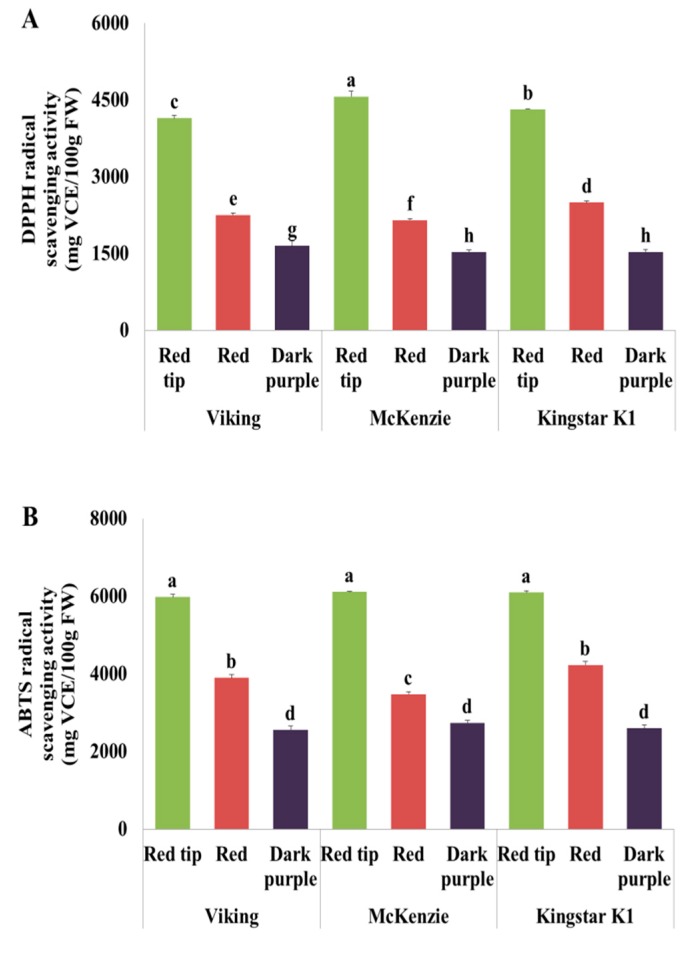
Antioxidant activities of *Aronia melanocarpa* fruits at different maturity stages: (**A**) DPPH radical scavenging; (**B**) ABTS radical scavenging. Vertical bars indicate standard deviation. Different letters are significant differences by Duncan’s multiple range test (*p* < 0.05).

**Table 1 foods-08-00598-t001:** Hunter L, a, b color of *Aronia melanocarpa* fruits at different maturity stages.

Cultivar	Maturity Stage	L*	a*	b*
Viking	Red tip	43.53 ± 1.99 ^b^	−7.74 ± 1.36 ^e^	12.17 ± 2.14 ^a^
Viking	Red	27.23 ± 1.35 ^e^	9.86 ± 0.31 ^a^	2.33 ± 1.20 ^d^
Viking	Dark purple	23.81 ± 1.03 ^g^	1.02 ± 0.31 ^c^	−1.84 ± 0.15 ^e^
McKenzie	Red tip	44.63 ± 2.66 ^a^	−6.34 ± 1.46 ^d^	9.83 ± 2.46 ^b^
McKenzie	Red	28.75 ± 1.57 ^d^	10.18 ± 1.85 ^a^	3.54 ± 1.34 ^c^
McKenzie	Dark purple	25.28 ± 0.74 ^f^	1.00 ± 0.36 ^c^	−2.42 ± 0.36 ^e^
Kingstar K1	Red tip	42.65 ± 1.17 ^c^	−6.77 ± 0.84 ^d^	9.90 ± 1.14 ^b^
Kingstar K1	Red	28.22 ± 1.56 ^d^	9.14 ± 1.51 ^b^	2.72 ± 1.44 ^d^
Kingstar K1	Dark purple	22.26 ± 0.72 ^h^	1.19 ± 0.44 ^c^	−1.78 ± 0.15 ^e^

Color measurement was expressed as lightness (L*), redness (a*), and yellowness (b*). Results are the mean values ± standard deviation from three measurements (*n* = 3); means in the same column with superscript with different letters (a, b, c, d, e, f, g, and h) are significantly different at *p* < 0.05.

**Table 2 foods-08-00598-t002:** Soluble solid contents (SSC), firmness, titratable acidities, pH and moisture content of *Aronia melanocarpa* fruits at different maturity stages.

Cultivar	Maturity Stage	SSC (°Brix)	Firmness (N/12 mmØ)	Titratable Acidity (%)	Malic Acid (mg/100 g)	pH	SSC/TA Ratio	Moisture Content (%)
Viking	Red tip	16.00 ± 1.00 ^b^	3.95 ± 0.34 ^b^	1.26 ± 0.04 ^b^	448.62 ± 6.50 ^e^	3.52 ± 0.06 ^de^	12.72 ± 0.87 ^b^	36.92 ± 1.03 ^d^
Viking	Red	12.33 ± 0.58 ^d^	2.05 ± 0.21 ^d^	1.30 ± 0.01 ^a^	658.29 ± 31.30 ^c^	3.47 ± 0.03 ^e^	9.48 ± 0.50 ^d^	41.08 ± 0.82 ^c^
Viking	Dark purple	13.66 ± 1.53 ^cd^	0.99 ± 0.23 ^f^	1.29 ± 0.01 ^a^	827.78 ± 42.63 ^a^	3.68 ± 0.06 ^c^	10.57 ± 1.23 ^cd^	54.87 ± 1.14 ^a^
McKenzie	Red tip	20.33 ± 1.53 ^a^	4.04 ± 0.18 ^ab^	1.28 ± 0.01 ^ab^	434.15 ± 16.78 ^e^	3.79 ± 0.02 ^b^	15.94 ± 1.29 ^a^	38.58 ± 1.28 ^cd^
McKenzie	Red	13.00 ± 1.00 ^cd^	2.26 ± 0.25 ^c^	1.29 ± 0.01 ^a^	669.78 ± 22.02 ^c^	3.58 ± 0.09 ^d^	10.09 ± 0.78 ^cd^	39.21 ± 0.87 ^cd^
McKenzie	Dark purple	14.67 ± 0.58 ^bc^	1.15 ± 0.20 ^e^	1.28 ± 0.01 ^ab^	815.00 ± 21.42 ^a^	4.01 ± 0.02 ^a^	11.50 ± 0.49 ^bc^	46.64 ± 1.70 ^b^
Kingstar K1	Red tip	20.67 ± 1.53 ^a^	4.09 ± 0.26 ^a^	1.29 ± 0.01 ^ab^	448.73 ± 16.77 ^e^	3.79 ± 0.02 ^b^	16.04 ± 1.20 ^a^	40.06 ± 0.69 ^c^
Kingstar K1	Red	13.00 ± 1.00 ^cd^	2.34 ± 0.27 ^c^	1.29 ± 0.01 ^a^	578.43 ± 29.32 ^d^	3.55 ± 0.03 ^de^	10.06 ± 0.88 ^cd^	38.44 ± 2.17 ^cd^
Kingstar K1	Dark purple	14.00 ± 1.00 ^bcd^	0.88 ± 0.15 ^f^	1.28 ± 0.01 ^ab^	758.33 ± 36.45 ^b^	3.81 ± 0.06 ^b^	10.90 ± 0.78 ^cd^	45.92 ± 2.69 ^b^

Results are the mean values ± standard deviation from three measurements (*n* = 3); means in the same column with superscript with different letters (a, b, c, d, e, and f) are significantly different at *p* < 0.05. SSC/TA ratio is calculated by SSC (°Brix)/titratable acidity (%).

**Table 3 foods-08-00598-t003:** Individual sugar contents of *Aronia melanocarpa* fruits at different maturity stages.

Cultivar	Maturity Stage	Fructose (mg/100 g FW)	Glucose (mg/100 g FW)	Sorbitol (mg/100 g FW)	Total Sum (mg/100 g FW)
Viking	Red tip	113.51 ± 33.48 ^f^	272.83 ± 16.64 ^f^	1343.97 ± 45.38 ^e^	1730.31
Viking	Red	841.90 ± 85.80 ^e^	972.32 ± 76.02 ^e^	1962.84 ± 66.08 ^d^	3777.05
Viking	Dark purple	2103.11 ± 194.48 ^b^	2366.75 ± 250.94 ^b^	3882.51 ± 536.95 ^b^	8352.37
McKenzie	Red tip	17.63 ± 11.53 ^f^	191.35 ± 8.13 ^f^	1333.54 ± 37.89 ^e^	1542.51
McKenzie	Red	1065.64 ± 108.32 ^d^	1145.51 ± 105.51 ^e^	2518.72 ± 323.36 ^c^	4729.86
McKenzie	Dark purple	2477.12 ± 74.65 ^a^	2677.70 ± 119.95 ^a^	4831.56 ± 129.08 ^a^	9986.37
Kingstar K1	Red tip	10.51 ± 14.92 ^f^	232.09 ± 21.57 ^f^	1432.02 ± 18.61 ^e^	1674.62
Kingstar K1	Red	1252.83 ± 108.20 ^c^	1405.74 ± 95.03 ^d^	2747.56 ± 210.53 ^c^	5406.13
Kingstar K1	Dark purple	1961.00 ± 110.10 ^b^	2080.48 ± 121.71 ^c^	3593.35 ± 173.51 ^b^	7634.84

Results are the mean values ± standard deviation from three measurements (*n* = 3); means in the same column with superscript with different letters (a, b, c, d, e, and f) are significantly different at *p* < 0.05. FW: fresh weight. FW: Fresh weight.

**Table 4 foods-08-00598-t004:** Polyphenol contents of *Aronia melanocarpa* fruits at different maturity stages.

Cultivar	Maturity Stage	Catechol (mg/100 g FW)	Chlorogenic Acid (mg/100 g FW)	Total Sum (mg/100 g FW)
Viking	Red tip	256.11 ± 19.27 ^c^	120.82 ± 7.91 ^b^	376.92
Viking	Red	134.18 ± 6.48 ^d^	72.31 ± 4.81 ^c^	206.49
Viking	Dark purple	95.05 ± 6.93 ^ef^	64.69 ± 3.09 ^cd^	159.74
McKenzie	Red tip	399.01 ± 22.22 ^a^	176.62 ± 15.57 ^a^	575.63
McKenzie	Red	118.02 ± 1.87 ^de^	68.75 ± 1.59 ^cd^	186.78
McKenzie	Dark purple	81.80 ± 3.50 ^fg^	56.72 ± 3.83 ^de^	138.53
Kingstar K1	Red tip	289.03 ± 10.88 ^b^	132.58 ± 5.39 ^b^	421.61
Kingstar K1	Red	118.33 ± 1.59 ^de^	72.87 ± 2.81 ^c^	191.20
Kingstar K1	Dark purple	70.69 ± 0.27 ^g^	49.76 ± 0.95 ^e^	120.45

Results are the mean values ± standard deviation from three measurements (*n* = 3); means in the same column with superscript with different letters (a, b, c, d, e, f, and g) are significantly different at *p* < 0.05.

**Table 5 foods-08-00598-t005:** Pearson correlation among physicochemical qualities, antioxidant compounds, and activities of *Aronia melanocarpa* fruits.

	a*	b*	SSC	Firmness	Titratable Acidity	Malic Acid	pH	Moisture Content	Fructose	Glucose	Sorbitol	Total Anthocyanin	Total Flavonoids	Total Phenolics	DPPH	ABTS	Catechol	Chlorogenic Acid
L*	−0.739 **	0.959 **	0.788 **	0.963 **	−0.305	−0.916 **	−0.071	−0.614 **	−0.885 **	−0.868 **	−0.798 **	−0.723 **	0.986 **	0.986 **	0.981 **	0.967 **	0.935 **	0.925 **
a*		−0.592 **	−0.792 **	−0.578 **	0.456 *	0.508 **	−0.384 *	0.068	0.441 *	0.406 *	0.335	0.124	−0.726 **	−0.757 **	−0.694 **	−0.622 **	−0.708 **	−0.706 **
b*			0.660 **	0.962 **	−0.191	−0.950 **	−0.267	−0.723 **	−0.936 **	−0.926 **	−0.884 **	−0.835 **	0.953 **	0.927 **	0.958 **	0.961 **	0.897 **	0.855 **
SSC				0.715 **	−0.461 *	−0.659 **	0.370	−0.292	−0.606 **	−0.576 **	−0.487 *	−0.384 *	0.805 **	0.837 **	0.790 **	0.731 **	0.830 **	0.820 **
Firmness					−0.192	−0.961 **	−0.212	−0.731 **	−0.954 **	−0.943 **	−0.889 **	−0.857 **	0.971 **	0.960 **	0.980 **	0.987 **	0.906 **	0.892 **
Titratable acidity						0.225	−0.429 *	0.080	0.037	0.028	−0.093	−0.038	−0.287	−0.286	−0.243	−0.179	−0.282	−0.272
Malic acid							0.276	0.791 **	0.941 **	0.933 **	0.891 **	0.893 **	−0.938 **	−0.909 **	−0.949 **	−0.971 **	−0.869 **	−0.853 **
pH								0.362	0.383 *	0.399 *	0.488 **	0.543 **	−0.087	−0.013	−0.126	−0.200	0.000	0.005
Moisture content									0.748 **	0.768 **	0.733 **	0.885 **	−0.629 **	−0.580 **	−0.642 **	−0.711 **	−0.535 **	−0.515 **
Fructose										0.998 **	0.981 **	0.911 **	−0.912 **	−0.883 **	−0.926 **	−0.942 **	−0.857 **	−0.835 **
Glucose											0.985 **	0.920 **	−0.895 **	−0.862 **	−0.909 **	−0.929 **	−0.837 **	−0.813 **
Sorbitol												0.907 **	−0.835 **	−0.794 **	−0.854 **	−0.877 **	−0.781 **	−0.758 **
Total anthocyanin													−0.753 **^z^	−0.705 **	−0.780 **	−0.839 **	−0.674 **	−0.655 **
Total flavonoids														0.991 **	0.997 **	0.979 **	0.961 **	0.949 **
Total phenolics															0.988 **	0.970 **	0.953 **	0.940 **
DPPH																0.987 **	0.954 **	0.943 **
ABTS																	0.917 **	0.903 **
Catechol																		0.995 **

Pearson correlation (*R*): **, significance at *p* < 0.01, *, significance at *p* < 0.05. DPPH: 2,2-diphenyl-1-picrylhydrazyl; ABTS: 2,2-azino-bis (3-ethylbenzothiazoline-6-sulfonic acid). L*: lightness; a*: redness; b*: yellowness.

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
