# Peer review of "Influence of Ripening Stage and Cultivar on Physicochemical Properties and Antioxidant Compositions of Aronia Grown in South Korea"

_foods, 2019, doi:10.3390/foods8120598_

Round 1

Reviewer 1 Report

I found the paper interesting. I think it is well written, well referenced and has enough scientific quality to be published in Foods. The data presented will be useful for scientist and future research. I recommend publishing the manuscript after minor revisions. I have some comments that need to be addressed by the authors:

I am missing method validation data (LOD, LOQ, stability, and so on). Please include. In the tables, please discard insignificant figures. Lines 227-233- I don’t think it is necessary to describe all data in the text, that is why is in the table. Get to point! Sugar content increase during the ripening process. Lines 267-272- ‘’Very high’’ Please, change and use a type of language more appropriate for a scientific article. Also put concentration. Besides, from lines 270 to 272, this statement seems to be out of the scope of the study and it is not supported by data presented here nor with references.

Reviewer 2 Report

GENERAL COMMENTS

Materials and methods should be improved with more details:

About experimental plan (agronomical conditions, number of samples per variety, number of independent samplings per variety at each harvest, number/weight of fruits per sample) Meteo climatic and soil characteristics of the site of study Analytical methods are not described in details (e.g. sample quantities/dilutions, number of measurements).

Results and discussion:

This part lacks discussion; it is mostly a repetition of results already reported in tables. It could be improved with some comments about the varietal differences (the principal aim of the work). You might also try to comment, explain, the broad changes in chemical composition of Aronia fruits you observed during maturation process under a physiological point of view, comparing also these results with those of other berries of the same family. You should improve the discussion comparing your results on the chemical composition of black chokeberry with more literature. Some recent reviews on this topic may help you to enhance your comparison (doi:10.3390/molecules24203710; https://doi.org/10.1016/j.tifs.2019.05.006). Comment the analysis in the same order you described them in materials and methods, with this way you make the paper easier to be read and understood.

Conclusions

There are many redundant sentences and concepts. What about the varietal influence on physicochemical properties and antioxidant composition o fruits? There are relevant differences between varieties? On the basis of you results (both for timing and quantities), can you say that one of them is better than the others?

English style can be improved. Some sentences are too large or hard to understand. It would be desirable a revision from an English native speaker.

SPECIFIC COMMENTS

Lines 48 – 50 and 54 – 56: the sentence is repeated, I would like to suggest removing the second one. Lines 60 – 71: please provide more details about the experimental plan. Could you describe the agronomic and pedo-climatic growing conditions? Were they the same for each cultivar? How many plants per cultivar did you consider? Did you make more than one independent sample per each harvest and each variety? So, which is the overall number of samples? How many fruits per sample did you harvest? The timing of fruit colour change was contemporary for all the three varieties? Lines 78 – 83: the method described is not clear. Could you provide more details? e.g. about the quantity of 80% ethanol per quantity fruit sample. What do you mean with “concentrated”? Did you evaporate only ethanol or what? What did you obtain finally? Line 91: how many fruits did you use to measure firmness? Lines 110 – 111: you should indicate the origin of standards, I suggest in paragraph 2.2, to be modified in “standards and chemical reagents”. Same thing for the standards mentioned in paragraphs 2.7, 2.9, 2.10, 2.11 Line 152: correct 30:970 with 30:70. Lines 170 – 175 statistical analysis: you made ANOVA analysis and Duncan’s test, but you should indicate how many repetitions of samples (of fruits at harvest for each variety) and how many repetitions for each measurement. Lines 208 – 220: in materials and methods you indicated the use of 8 standards for organic acids analysis. Why in paragraph 3.3 you describe only malic acid? What about the other organic acids? Table 2: the standard deviation and significance letter of Dark purple Kingstar K1 SSC value is missing. Lines 237 – 240 and 240 – 242 the two sentences are in contradiction but citation is the same, something is wrong. Lines 243 – 272: according to materials and methods you expected to find 14 phenolic compounds (according to the number of standards you used). However, in results and discussion (paragraph 3.5) you discuss only catechol (did you find other works reporting this molecule in Aronia fruits or extracts?) and chlorogenic acid as the main phenolic compounds of Aronia (data in table 4); what about the other compounds? Did you detect them? If yes, could you provide the data? Could you also provide a representative HPLC chromatogram? Figure 2 and 3: you may join them, in a single figure with three charts. You will improve the text flow. Tables and Figures: you should move up the tables and figures close to the respective paragraph. Lines 353 – 378 paragraph 3.11: you can improve this part with more detailed statistical analyses. You can apply several types of regression models (e.g. stepwise linear regression model, PLS, OPLS) to identify the chemical compounds, among the ones you identified, that mostly contribute to the antioxidant activity of Aronia fruit extracts.

Reviewer 3 Report

The paper entitled “Influence of ripening stage and cultivar on physicochemical properties and antioxidant compositions of Aronia grown in South Korea” investigated the changes of chemical composition of the juice of aronia fruit over maturation. In particular, the authors focused their attention on sugars and antioxidant compounds.

Some points to be improved

Anthocyanins are the main compounds in red berries and particularly in Aronia berries that are widely used as natural pigments. However, despite the importance of anthocyanin compounds, the authors do not discuss in the abstract the changes of anthocyanin content during maturation. Similarly, in the keywords, the word anthocyanins is not listed. The authors should change the word correlation, that it does not make sense, with anthocyanins. The objectives of the research are too generic and not well defined (see pag 2 line 53). The results of this paper focus on the changes of antioxidant compounds and sugars during maturation of aronia fruit and not on the possible uses of aronia at different stages of maturity. Moreover, the last objective  (pag 2 line 56-57) it is not a goal easily supported by the results presented. The authors should modify the objectives of the manuscript in order to be more precise. In materials and methods, some points should be improved.

        - pag. 2  plant material section: please provide the amount of berries used for the analyses

        - pag. 2 line 88: was the color measured three times always in the same position of the fruit? Please specify.

        - pag 3, line 100: please specify for how many time the berries were left at 60 °C.

        - pag 4 lines 159-163. Please specify how was the solution of DPPH prepared (please specify the solvent used).

        - pag 5 statistical analysis section. Please specify which kind of ANOVA was used.

In the result and discussion section, it is not necessary to repeat in the text what is already evident in the tables.

Round 2

Reviewer 2 Report

The quality of the manuscript has improved. However, it still needs more improvements, principally on discussion of results, still poor.

Paragraph 2.1: please include the information you provided in the answer to point 8: “There were total 27 samples because of 3 cultivars X 3 maturities X 3 replicates. We collected 3 kg for each sample.” Line 170 please modify “3% acetic acid (acetic acid:distilled water = 30:970)” with “Acetic acid (3%)” Statistical analysis: one-way ANOVA can be performed only to evaluate a single factor. In this case you can evaluate the effect of ripening on each individual cultivar and, separately, the cultivar effect per each ripening stage. Please correct the one-way ANOVA and, consequently, Duncan’s multiple range tests. Paragraphs 3.3 and 3.8: please include in the text the information about the other organic acids and phenolic compounds you tested. Try to explain why you could not quantify them differently to other authors. You also should consider, if possible, repeating analyses with concentrated solutions. Conclusions: please include some comments about the “cultivar effect” on antioxidant composition.
